# Multiple Actions of Telomerase Reverse Transcriptase in Cell Death Regulation

**DOI:** 10.3390/biomedicines11041091

**Published:** 2023-04-04

**Authors:** Anastasia I. Palamarchuk, Elena I. Kovalenko, Maria A. Streltsova

**Affiliations:** Shemyakin & Ovchinnikov Institute of Bioorganic Chemistry, Russian Academy of Sciences, Ul. Miklukho-Maklaya 16/10, 117997 Moscow, Russia

**Keywords:** telomerase, TERT, telomerase reverse transcriptase, telomeres, gene regulation, stress, cell death, apoptosis, survival

## Abstract

Telomerase reverse transcriptase (TERT), a core part of telomerase, has been known for a long time only for its telomere lengthening function by reverse transcription of RNA template. Currently, TERT is considered as an intriguing link between multiple signaling pathways. The diverse intracellular localization of TERT corresponds to a wide range of functional activities. In addition to the canonical function of protecting chromosome ends, TERT by itself or as a part of the telomerase complex participates in cell stress responses, gene regulation and mitochondria functioning. Upregulation of TERT expression and increased telomerase activity in cancer and somatic cells relate to improved survival and persistence of such cells. In this review, we summarize the data for a comprehensive understanding of the role of TERT in cell death regulation, with a focus on the interaction of TERT with signaling pathways involved in cell survival and stress response.

## 1. Introduction

Telomerase is constitutively expressed in stem cells, including progenitor cells of skin, intestine and hematopoietic niches. It is temporarily induced in a number of proliferating cells, for example, in lymphocytes upon stimulation [1]. In general, telomerase activity diminishes in normal somatic cells simultaneously with their differentiation, whereas in 85–90% of malignant neoplasms telomerase reactivation is observed [2]. Such reactivation is associated with an increased survival of cancer cells [2]. Telomerase has been known since 1985 for its ability to extend telomeric repeats on chromosome ends (canonical function) to ensure the stability of genetic material [3], but the identification of other, non-telomeric (non-canonical) functions continues to this day [4].

Tumor progression, accompanied by an increased expression of telomerase, is often associated with an activation of various signaling pathways, involved in energy metabolism, survival and cell cycle progression [5]. Therefore, the role of telomerase in the regulation of cell fate (death or survival), both cancer and somatic, including lymphocytes, becomes especially interesting. A relationship was shown between the level of telomerase activity and increased resistance to therapeutic treatment along with a subsequent poor prognosis for cancer patients [6,7,8], partly because telomerase supports cell survival by suppressing apoptotic signaling [9]. Hence, telomerase mediates many protective functions which will be discussed further.

## 2. Telomere Structure and the Role in DNA Damage Response

Telomeres are protective structures at the ends of chromosomes represented by short repeating oligonucleotide sequences (5′-TTAGGG_n_) in complex with proteins called shelterins, which are necessary to maintain the stability of telomeres [10]. In each division cycle, telomeric ends are shortened due to the “end replication problem”, which occurs because of incomplete replication of the lagging strand [11,12,13] and results in the limited proliferative potential of somatic cells [14]. Telomeres protect genome stability, preventing chromosome end fusion and undesirable recombinations. In the absence of a protective mechanism, the ends of linear chromosomes would mimic a unilateral break of double-stranded (ds) DNA and might trigger a cellular DNA damage reaction, the result of which would be either a replication arrest or cell death [10,15]. The shelterin proteins are involved in the stabilization of the telomere structure, t-loop formation and interaction with DNA repair pathways. Human shelterin complex includes TRF1, TRF2, Rap1, TIN2, TPP1 and POT1 [16]. These proteins specifically interact with both dsDNA and 50–400 nucleotide 3′ protrusion of single-stranded (ss) telomeric DNA, contributing to the t-loop formation, which makes it possible to distinguish the telomeric ends from the rest of the double-stranded DNA breaks.

Dysfunction of the normal telomere structure initiates DNA damage response (DDR) which enables the activation of ataxia telangiectasia mutated kinase (ATM) and ataxia telangiectasia and Rad3-related (ATR) signaling pathways that play a central role in maintaining the genome integrity. Simultaneous signaling of both ATM and ATR as a result of their cross-talk leads to the activation of a number of effector proteins, including checkpoint kinases 1 and 2 (CHK1, CHK2) and p53, which regulate cell cycle progression, apoptosis and DNA repair [17,18]. Inhibition or removal of the shelterin protein TRF2 leads to the activation of the ATM kinase pathway, followed by up-regulation of p53 [10,16,19]. Apparently, the t-loop structure itself is capable of preventing DDR by increasing the level of DNA compactization by modification of core histones and their interaction with other factors [10,16,19]. Unlike the ATM-mediated pathway, which is triggered in response to the dsDNA break, the ATR pathway is activated when ssDNA is recognized. The constitutive 3’ protrusion of mammalian telomeres is long enough for binding by replication protein A (RPA) proteins that are used to bind ssDNA and subsequently activate ATR on unprotected telomeres. Thus, the shelterin protein POT1 suppresses ATR signaling, possibly by blocking the binding of telomeric ssDNA to RPA [10,20].

### 2.1. Canonical Function of Telomerase

Telomere homeostasis in normal somatic cells is supported by the activity of the telomerase holoenzyme. However, telomerase reverse transcriptase activity is normally downregulated in differentiated somatic cells to prevent their uncontrolled proliferation or malignant transformation. Nevertheless, in a number of cases telomerase reactivation is observed in physiological conditions, for example, in lymphocytes stimulated for robust proliferation [21,22,23]. The core of telomerase holoenzyme is represented by two key components necessary for its reverse transcriptase activity: telomerase RNA component (TERC) and telomerase reverse transcriptase (TERT). The canonical function of telomerase corresponds to the DNA elongation at the telomeric ends of chromosomes by reverse transcription of oligonucleotide sequences of the telomerase RNA template [15]. TERT expression is accurately regulated in various cells in order to avoid cancer cell transformation, whereas TERC is constitutively expressed in mammalian cells [23,24,25]. Most transformed cells use telomerase to lengthen and preserve telomeres, however, about 4–11% of cancers use a pathway based on homologous recombination called alternative lengthening of telomeres (ALT) [26].

### 2.2. Role of Telomerase in DNA Damage Response

Telomerase is not only directly involved in the genome stability by forming protective telomeric ends of chromosomes. An increased TERT cell level was shown to be associated with the enhanced functioning of DNA repair machinery, namely, with reduced spontaneous DNA damage and improved DNA repair kinetics. However, no evidence of direct interaction of TERT with proteins of DNA repair machinery was observed. Likely, TERT participates in the modulation of their expression [27]. Moreover, Masuomi et al. have shown that an increase or decrease in the TERT level significantly affects the chromatin structure and, accordingly, the radiosensitivity of the cells. They established a direct link between the mechanisms that maintain structure and integrity of telomeres and those that detect and repair breaks all over the chromosome [28]. Indeed, the TRF2 protein, which normally binds telomeres, appeared to be temporarily localized in DNA breaks [29] while many proteins involved in DDR were associated with telomeres [30]. 

Thus, telomerase adds telomeric repeats and thereby protects the integrity of a complex deoxyribonucleoprotein structure that masks the chromosome ends, which otherwise can be recognized as double-stranded DNA breaks by DNA damage response machinery. Furthermore, telomerase indirectly restrains cell death induction through an activation of DDR via ATM and ATR signaling pathways to maintain the stability of the genome integrity. In addition, it is suggested that there is a relationship between the processes of DNA repair and telomere elongation which also contributes to the cell survival under stress conditions (Figure 1).

## 3. Non-Canonical Functions of Telomerase Reverse Transcriptase (TERT) 

In recent years, increasing data appears on TERT, one of main telomerase components, to perform so-called non-canonical functions that are independent of telomerase activity. Various intracellular localizations (nucleus [31,32], cytoplasm [33,34], mitochondria [35,36,37]) of the telomerase catalytic subunit correspond to a wide range of functional activities [38,39]. It became evident that TERT is involved in the regulation of various signal transduction pathways and even in gene expression [32,40,41]. TERT mediates protection from an excessive production of reactive oxygen species (ROS) or cytosol acidification by mitochondria [37,42], endoplasmic reticulum (ER) stress [43] and cell death [4,5,44,45,46]. TERT is involved in the regulation of metabolism [47], autophagy [48] and maintenance of cells’ Red/Ox potential [49]. TERT was also shown to affect the resistance to apoptosis. TERT suppression causes apoptosis or increases the sensitivity of cancer cells to apoptotic stimuli ex vivo and in vivo [41,50], regardless of the telomeric function [51]. The transduction of the *TERT* gene or a catalytically inactive mutant protects various tumor cell lines from apoptosis, illustrating that the observed phenomena do not depend on the canonical functions of TERT [38,52].

The non-canonical functions of telomerase are extensive and multifaceted. Meanwhile, overall, telomerase reduces the risk of cell death induction by affecting wide range of processes involved in regulation of cell survival and overcoming stressful conditions. Later in this article, we discuss in detail telomerase involvement in a fine balance between cell death and survival.

## 4. Principal Intracellular Molecular Pathways of the Regulated Cell Death

Regulated cell death (RCD) is a genetically programmed mechanism necessary for the normal development of multicellular (and a number of unicellular) organisms, allowing timely removal of damaged, altered and potentially dangerous cells. RCD is a complex molecular system that is finely regulated by external and internal factors. Thus, there are two different but interrelated pathways, internal and external, leading to cell death [53]. The first, intrinsic apoptosis, is a pathway of cell death that develops in response to cell stress: ER stress, DNA damage or ROS overload. Such cell stress may be induced by infections, cytokine and growth factor deprivation, or other environmental factors [54,55,56,57]. The second, the external pathway of apoptosis, is triggered in response to the binding of death receptors to the corresponding ligands [53,58]. In addition, it is worth noting a form of cell death caused by exogenous mediators. For example, granzyme B, secreted by cytotoxic T lymphocytes and NK cells, enters a target cell through the immunological synapse during killing [58]. All these pathways ultimately induce a cascade of caspase activation and subsequent cell death; however, fine regulation of signaling at several levels makes it possible to slow down or even inhibit this process [53].

### 4.1. Intrinsic Apoptosis

The key stage of intrinsic apoptosis is the mitochondrial outer membrane permeabilization (MOMP) which is controlled by a balance of pro- and anti-apoptotic proteins of the BCL-2 family [59]. This family consists of three subgroups of proteins sharing the BCL-2-homology (BH) domain. Among them, pro-apoptotic BH3-only initiators and the membrane permeabilizing effectors, and anti-apoptotic guardians are distinguished (Figure 2).

Pro-apoptotic BH3-only initiators, BH3-interacting domain death agonist (Bid), p53 upregulated modulator of apoptosis (Puma), BCL-2-interacting mediator of cell death (Bim) and phorbol-12-myristate-13-acetate-induced protein 1 (Noxa) are mediators of cellular response to stress. Activated post-transcriptionally (such as Bid, for example) or by enhanced transcription (such as Puma, Bim and Noxa), they promote the apoptosis progression through direct interaction with the death effectors BCL-2-associated X protein (Bax) and BCL-2 antagonist/killer (Bak). Bax normally circulates between the outer mitochondria membrane (OMM) and the cytosol. In turn, Bak is located on the OMM and interacts with voltage-dependent anion channel (VDAC2). Under the apoptosis stimulation, Bax and Bak acquire the ability to oligomerize and then to form pores in the OMM [53,60]. Anti-apoptotic guardians, such as B-cell lymphoma 2 (Bcl-2), B-cell lymphoma-extralarge (Bcl-X_L_), myeloid cell leukemia-1 (Mcl-1) and BCL-2-like protein 2 (Bcl-W) proteins, act oppositely to pro-apoptotic proteins. Most anti-apoptotic members of the BCL-2 family inhibit Bax and Bak, preventing their oligomerization either by their direct physical isolation from OMM, or indirectly through sequestration of the BH3-only activators. The anti-apoptotic functions of the BCL-2 guardians can be blocked by their binding to a group of pro-apoptotic BH3-only proteins of the BCL-2 family known as “sensitizers”: BCL-2-associated agonist of cell death (Bad), BCL-2 modifying factor (Bmf) and BCL-2-interacting protein (Hrk). These proteins are capable of provoking MOMP without interacting with Bax and Bak. Additionally, Bim, Bid and Puma are able to bind all anti-apoptotic BCL-2s, whereas others predominantly inhibit individual targets: Bad interacts Bcl-2, Bcl-X_L_ and Bcl-W; Noxa blocks Mcl-1; and Hrk inhibits Bcl-X_L_ [53,61]. As a result, permeabilization of the outer mitochondrial membrane leads to the release of cytochrome c (cyt c) and second mitochondria-derived activator of caspases/Diablo homolog (SMAC/DIABLO) to the cytoplasm, which initiates the assembly of the apoptosome with the participation of cyt c, apoptotic protease-activating factor 1 (APAF1) and procaspase 9. Next, activated caspase 9 cleaves and triggers the effector caspases 3 and 7. SMAC/DIABLO, in turn, binds inhibitors of apoptosis (IAPs), which normally block the activity of caspases. Interestingly, X-linked inhibitor of apoptosis protein (XIAP) is the only IAP that directly physically blocks caspases. The caspase cascade activation leads to cell protein proteolysis and subsequent cell death accompanied by DNA fragmentation and phosphatidylserine exposure [62].

### 4.2. Extrinsic Apoptosis

Extrinsic apoptosis is triggered in response to the binding of a death receptor to its ligand, which stimulates the assembly of the intracellular death-inducing signaling complex (DISC) and the caspase 8 activation. Cellular FLICE-like inhibitory proteins (c-FLIP) modulate signal transduction and pro-caspase cleavage and thereby regulate cell death induction. There are three isoforms of c-FLIP: c-FLIP_S_, c-FLIR_R_ and c-FLIP_L_. Whereas c-FLIP_S_ inhibits signal transduction from death receptors, c-FLIP_L_ and c-FLIP_R_ facilitate cell death signaling transduction [63]. The further apoptotic cascade could proceed in two ways. In so-called “type I cells” (for example, thymocytes and mature lymphocytes), the activated caspase 8 directly proteolytically cleaves the effector caspases 3 and 7 [64,65]. On the contrary, in “type II cells” (for example, hepatocytes, pancreatic β-cells and most cancer cells), in which the activation of caspase 3 and caspase 7 is restrained by XIAP, external apoptosis requires caspase-8-mediated proteolytic cleavage of Bid to tBid. tBid, in turn, activates Bax and Bak, which induce MOMP. This process is followed by the release of SMAC/DIABLO and activation of caspase 9 [53,66].

### 4.3. Cell Death Induction by Granzymes—The Lytic Mechanism of Immune Cells

Since significant antitumor protection is provided by immune cells, such as NK cells and cytolytic T lymphocytes, capable of eliminating altered cells both by releasing lytic granules and by triggering extrinsic apoptosis through the death receptors [58], it is interesting to consider the mechanisms of death induction in tumor cells overexpressed telomerase by components of lytic granules.

Perforin, granulysin and granzymes are the main components of lytic granules. Perforin, due to its pore-forming ability, facilitates the delivery of lytic content into the target cells, and granzymes penetrate and finely eliminate inappropriate cells [58,67]. Granzymes are the family of closely related serine proteases that are expressed mainly in cytotoxic T cells and NK cells [68]. Within the family, granzyme B has been studied in the most detail. It activates the caspase pathway of apoptosis by cleavage of caspases 3 and 7 which leads to destruction of many intracellular substrates [68,69]. Granzyme B was also described as initiating apoptosis through Bid cleavage and formation of truncated tBid that mediates MOMP by interaction with Bax and Bak [70]. However, granzyme B-induced apoptosis can be inhibited by overexpression of the anti-apoptotic protein Bcl-2 [71].

Granzyme A can rapidly induce cell death in a manner other than caspase activation and mitochondrial permeabilization. In fact, large DNA fragments, which are not detected by conventional apoptosis tests, were observed under granzyme A treatment [58,72]. Within minutes granzyme A is able to disrupt the ER and mitochondria functionality and lead to the accumulation of intracellular ROS [58,73]. The caspase-independent cell death has been also shown for other granzymes, such as K, M and H [58,67,68].

Thus, granzymes are able to destroy multiple cellular components and induce cell death, either directly through the activation of caspase-dependent apoptosis with the disruption of mitochondrial integrity resulting in the release of cyt c, or by triggering cell stress signaling caused by multiple destructions of intracellular contents.

## 5. TERT Directly or Indirectly Affects the Expression of Genes of Various Signaling Pathways

It is now well known that TERT acts as a transcriptional (co-) factor for regulation of gene expression in a number of signaling pathways by a direct binding to promoter regions or through modulation of the activity of other transcription factors [46,74,75]. The increased expression of the *TERT* gene, which significantly improves cancer cell survival, is associated with an altered regulation of a number of apoptotic signaling proteins. It has been shown that the telomerase catalytic activity may affect the cell ratio of Bax/Bcl-2 factors [76,77]. Zhang et al. demonstrated that the TERT expression counteracts apoptosis by tuning of Bcl-2, Bcl-X_L_, Bax and caspase 3 levels in osteosarcoma cells. The expression of Bcl-2 and Bcl-X_L_ proteins was significantly upregulated, and Bax levels were reduced in cells transfected with wild-type TERT and catalytically-inactive TERT while opposite expression patterns were observed in TERT-siRNA-transfected cells [49]. Moreover, it has been shown that the promoter of the *TERT* gene itself is targeted by a number of transcription factors corresponding to the main signaling pathways, such as NF-κB, c-Myc, β-catenin and STAT3 [4,74,78,79], which act as positive regulators of the *TERT* gene, and p53, which inhibits *TERT* transcriptional activity [46,52]. The role of telomerase in relation to the main intracellular signaling pathways, such as Wnt/β-catenine, c-Myc, NF-kB and p53, at the transcriptional level, will be further considered in more detail.

### 5.1. TERT Participates in the Regulation of the Wnt/β-Catenine and c-Myc Pathways 

In 2008, it was shown that the genes regulated by TERT appear to be similar to the genes regulated by c-Myc and Wnt, two factors associated with cell stemness, differentiation and cancer. Thus, it turned out that TERT is able to convergently influence the development program of progenitor cells through the pathways typical for c-Myc and Wnt [31].

TERT is known to be a direct modulator of the Wnt/β-catenin pathway. In the cytoplasm, TERT interacts with BRG1 (chromatin remodeling protein) and is transported to the nucleus where it occupies promoter regions of the Wnt/β-catenin target genes [32], including *CCND1* (cyclin D1 gene) and *c-MYC* [74,80]. Interestingly, telomerase is also involved in the stabilization of c-Myc, while c-Myc acts as a positive regulator of *TERT* expression, thus creating a positive feedback loop [81]. Moreover, *TERT* appears to be a target gene for β-catenin in a complex with transcription factor 4 (TCF4) and Kruppel-like factor 4 (Klf4) [75,82]. The activity of β-catenin also depends on intracellular processes. Metabolic stress can lead to the activation of GSK3ß kinase and the inhibition of its antagonistic kinase Akt [83]. In turn, activated GSK3ß is able to degrade β-catenin [84] and to activate the pro-apoptotic protein Bax, which is normally blocked by the Akt kinase. Synthesis and degradation of Akt is also regulated by the activity of β-catenin [83].

Thus, TERT contributes to cell survival along the Wnt/β-catenin pathway, facilitating the transcription of the pathway target genes. In addition, a role of telomerase in the protection against cellular stress is suggested which consists of preventing the degradation of β-catenin and the activation of Bax.

### 5.2. TERT Is Involved in NF-κB Signalling Pathway

TERT forms a positive feedback loop with the NF-κB signaling pathway. It is known that the transcription factor NF-κB belongs to the positive regulators of *TERT* expression, while in the cytosol, TERT or the TERT+TERC complex can form TERT–NF-κB subunit p65 complex, which is able to migrate to the nucleus and regulate the expression of a wide range of NF-κB target genes [40]. One of the important functions of NF-κB is the induction of the expression of pro-survival genes. For example, NF-κB increases the expression of anti-apoptotic BCL-2 family members (Bcl-2 and, in particular, Bcl-X_L_). It also stimulates c-FLIP_S_ expression and so interferes with induction of external apoptosis from death receptors [85]. It is noteworthy that Bcl-2 overexpression increases both the catalytic activity and the expression of TERT due to the ability of Bcl-2 to enhance basal transcriptional activity of NF-κB [86,87]. NF-κB boosts the expression of caspase inhibitors: c-IAPs and XIAP [88]. Moreover, activation of NF-κB is associated with reduced expression of pro-apoptotic factors, such as Bax [89]. In addition to the abovementioned transcriptional upregulation of anti-apoptotic factors and downregulation of pro-apoptotic factors, telomerase can mediate protection from ER stress. Enhanced expression of TERT was observed within 1 h of ER stress induction, while apoptosis proceeded in cells with reduced TERT levels. Such TERT-mediated protection through its upregulation was associated with the translocation of the p65 NF-κB transcription factor into the nucleus [43].

Thus, TERT, through interaction with the NF-κB signaling pathway, is able to enhance cell survival in two ways: to increase the production of anti-apoptotic factors and to reduce the level of pro-apoptotic ones. This mechanism apparently is based on the principle of positive feedback, but this hypothesis still requires additional confirmation.

It has been described that the level of telomerase catalytic activity is associated not only with TERT regulation at the transcriptional level by NF-κB, but also with post-transcriptional regulation, namely the phosphorylation of TERT by Akt kinase. For example, in multiple myeloma cells, IL-6, one of the key inflammatory factors, and insulin-like growth factor 1 (IGF1) increased telomerase activity without changing the level of TERT expression. This has been shown to be related to the signaling through PI3K/Akt/NF-κB [74,90]. Hence, the members of the NF-κB pathway are able to regulate telomerase production and functional activity at distinct levels.

### 5.3. TERT Is Modulated by p53 Signaling

P53, a well-known human tumor suppressor located at the crossroads of many signaling pathways, is involved in the regulation of cellular response to stress and damaging factors. Depending on the upstream signals, p53 alters the transcription of a number of apoptotic factors [91]. Activation of p53 is associated with a decrease in TERT expression [52] simultaneously with a decrease in telomerase activity [92]. Several genes of pro-apoptotic factors of the BCL-2 family, including *BAX* [93], *BBC3* (BCL2 Binding Component 3; Puma-coding gene) [94] and *PMAIP1* (Phorbol-12-Myristate-13-Acetate-Induced Protein 1; Noxa-coding gene) [95], as well as *APAF1* [96] and genes of the TNF family receptors [97], are activated upon p53 induction [98].

There is a crosstalk between p53 and NF-κB. The tumor suppressor p53 and NF-κB are two principal transcription factors in the regulation of cellular survival under stress conditions and death receptor signaling. They regulate the expression of multiple apoptotic genes, but in general, p53 promotes cell death, while NF-κB inhibits it [99,100]. Thus, taking into account the existence of mutual regulation between p53, NF-κB and TERT, it can be concluded: (1) p53 suppresses the expression and transcriptase activity of TERT; (2) TERT forms a positive feedback loop with NF-κB; (3) activation of NF-κB stimulates the expression of *TERT* gene [40,86,87]; (4) NF-κB, in turn, negatively affects the stability of p53 through the degradation by E3 ubiquitin ligase mouse double minute 2 homologue (MDM2) [99,101].

There is also crosstalk between p53 and Wnt signaling [102]. A number of studies have shown that overexpression of β-catenin contributes to the accumulation of active p53 [103,104]. On the other hand, an increase in the p53 activity can induce β-catenin degradation, via GSK3β [105] and downregulation of TCF4 [106], a factor that forms a complex with β-catenin to upregulate *TERT* expression [75].

Thus, telomerase is associated with several major intracellular signaling pathways (Figure 3), which not only affect the production and functional activity of telomerase in different ways, but TERT itself acts like a transcription factor that can tune this process. These pathways are associated with the regulation of cell survival and their resistance to damaging and stressful factors which allows us to consider telomerase as a linking component in the mediating resistance to cell death.

## 6. The Role of TERT in the Regulation of Apoptosis in the Cytoplasm and Mitochondria

A number of studies have shown that cells with an increased level of TERT have greater resistance to external damaging factors and are more protected from cell death [8,33,35,41,49,50,52,77,107]. Such stability, according to recent data, is due not only to the intranuclear activity of TERT (lengthening of telomeric ends and regulation of gene expression), but also due to the its cytoplasmic function: cell death regulation through direct interaction with mitochondria and proteins of the apoptotic pathway. Besides, cell stress-induced TERT export from the nucleus was described [33,49,50]. Even more, the protective properties of TERT have been associated with the suppression of the mitochondrial pathway of apoptosis [35,50]. The Jin et al. group revealed that TERT catalytic subunit is capable of binding pro- and anti-apoptotic proteins, such as Bcl-X_L_, Mcl-1 and Bad, with its BH3-like motif, and so directly regulating the apoptosis induction. However, no data on TERT-mediated modulation of Bcl-X_L_/Bax complex interactions were obtained. TERT was also shown to bind Beclin-1 (BECN1), a autophagic cascade factor [44]. It was found that TERT can counteract the apoptosis progression by reducing the activation level of caspases 3, 8 and 9 in response to the stimulation of both internal and external pathways of apoptosis. Such resistance to apoptosis of cells overexpressing TERT was associated with a decrease in pro-apoptotic factors Bax and Bak and an increase in Bcl-2 levels along with unchanged IAPs and XIAP levels in the cells. Significantly increased levels of the phosphorylated form of the c-Jun N-terminal kinase (p-jnk) and low levels of tBid have also been described in cells overexpressing TERT. Moreover, p-jnk inhibition restored caspase activity and tBid levels and thus stimulated apoptosis [77]. In addition to intracellular regulation and pro-survival action, TERT, according to accumulating data, is able to regulate many pro- and anti-apoptotic factors both by directly binding them in the cytoplasm and by indirect modulation of their functionality through phosphorylation.

### TERT Protects Mitochondria Functionality

About 10–20% of total TERT is localized in the mitochondria, both in normal and cancer cells. TERT has an N-terminal mitochondrial targeting signal that could be phosphorylated under stress conditions and so direct TERT into mitochondria [38,42,108,109]. There, telomerase participates in mitochondrial DNA (mtDNA) repair, regulates the activity of respiratory chain and reduces ROS production [36,39,110]. On the other hand, Santos et al. showed that cells with elevated TERT levels exhibited greater H_2_O_2_-induced mtDNA damage, leading to loss of mitochondria membrane potential (δΨ) and subsequent induction of apoptosis [42,109,111]. Meanwhile, the Zhang et al. group showed that tumor cells with increased TERT expression are more resistant to the pro-apoptotic action of cisplastin, due to improved mitochondrial functioning and decreased intracellular ROS [49]. In another work, it was shown that the protective effect of TERT on the mtDNA integrity was accompanied by a decrease in the mitochondrial ROS level, along with increased δΨ both under acute and chronic oxidative stress. In summary, this indicates an improvement in the mitochondria functioning in cells overexpressing TERT. Such cells demonstrated enhanced resistance to the cell death induction under stress conditions. Cell destruction was also slowed down and accompanied with delayed retrograde response [112]. The retrograde response has been described to be a serious reprogramming of nuclear gene expression patterns, including genes involved in metabolism, stress response, and growth signaling, as a result of mitochondrial dysfunction and Ca^2+^-dependent signaling [113,114]. It was shown that the mitochondrial membrane potential was more stable in cells with overexpression of TERT [50]. In addition, TERT, localized in the mitochondrial matrix, binds to mitochondrial DNA in regions encoding genes of the mitochondrial respiratory chain and regulates their expression [36,115]. TERT binding to mitochondrial DNA protects against damage caused by ethidium bromide. TERT increases the overall activity of the respiratory chain which is most pronounced in complex I (NADH dehydrogenase) and depends on the activity of the enzymatic activity of telomerase reverse transcriptase. Moreover, mitochondrial reactive oxygen species increase after the TERT level drop caused by the TERT inactivation with siRNA or shRNA [6,7,35,37,115]. It is noteworthy that TERT is suggested to be involved in mtDNA repair by the reverse transcription. Interestingly, TERT is unable to use TERC since it is absent in mitochondria, but mitochondrial tRNAs can serve as a template for reverse transcription. Furthermore, lack of mitochondrial TERT leads to mitochondrial dysfunction which is possibly related to the activity of reverse transcriptase in the organelle [36].

Thus, despite the slightly contradictory data obtained by the Santos et al. group favoring the pro-apoptotic properties of TERT, there is still more data revealing TERT as a factor in cell protection at the mitochondrial level. TERT is able to maintain the functional activity of mitochondria, participate in the mtDNA repair, and thus prevent cell death from mitochondrial dysfunction.

The presence of NF-κB and p53 together with TERT in mitochondria is also intriguing, given their complex reciprocal regulation in the nucleus. Surprisingly, the NF-κB pathway proteins, such as nuclear factor of kappa light polypeptide b-cell enhancer gene alpha inhibitor (IκBα) and p65 subunits have been found in mitochondria. It was shown that the p50 and p65 NF-κB subunits bind to mtDNA and thus regulate the expression of electron transport chain (ETC) genes in mitochondria [116,117,118].

IkBα is also localized on the outer mitochondrial membrane to inhibit apoptosis. Such suppressive effect is especially pronounced in tumor cells with constitutively active NF-κB, which accumulates a large amount of mitochondrial IkBα, an NF-κB target gene [119]. In mitochondria, IkBα reduces the apoptosis induction through stabilization of the complex between VDAC1 and hexokinase II (HKII), thereby preventing the recruitment of Bax to VDAC1 and the cyt c release [120,121]. Both TERT and TERC regulate HKII expression. Overexpression of TERT or TERC simultaneously enhanced HKII promoter activity and mRNA expression [122]. Thus, an increase in the TERT expression provides another mechanism of anti-apoptotic protection via stimulation of NF-κB target genes, including the inhibitor IkBα, which, binding to mitochondria, stabilizes the complex of VDAC1 and hexokinase II (HKII). Moreover, recruitment of HKII to mitochondria was associated with Akt signaling activity. Akt has previously been shown to phosphorylate and inhibit Bax [123]. 

P53, also found in mitochondria, has a different, transcription-independent, pro-apoptotic effect, which includes direct interactions between p53 and MOMP inducers at the mitochondrial level. It has been shown that part of the stress-induced p53 protein is rapidly translocated into mitochondria in response to stress. Thus, p53 can interact with Bcl-2, Bcl-X_L_, Bax or Bak, inducing MOMP [124]. However, it might be interesting to elucidate whether there is any interaction or mutual regulation between TERT and p53 on the mitochondrial level. 

Besides interaction with TERC, TERT forms a separate ribonucleoprotein complex with the RNA component of mitochondrial-RNA-processing endoribonuclease (RMRP). The complex exhibits RNA-dependent RNA polymerase activity with an ability to produce dsRNA. Thus, TERT-RMRP produces dsRNAs where RMRP is a matrix for polymerase reaction. Obtained dsRNAs are processed by Dicer into 22 bp dsRNAs and then loaded into Ago2. It confirms that these short RNAs represent endogenous siRNAs. Thus, possibly, TERT is able to regulate the expression of some mitochondrial genes [125].

Altogether, TERT improves mitochondria protection in different ways. In particular, TERT participates in regulation of normal mitochondrial functioning, modulation of its gene expression and mtDNA integrity.

## 7. TERT Protects Cells from ER Stress 

ER stress temporarily activates TERT expression. It is important to note that TERT inhibition sensitizes cells to apoptosis under ER stress [126], whereas an increased expression of TERT reduces ER stress-induced cell death regardless of the catalytic activity of enzyme or DNA damage signaling [43,127]. TERT exerts the anti-apoptotic effect at an early stage of the cell death process, preceding mitochondrial dysfunction and caspase activation [128]. TERT knockdown leads to a change in the regulation of several key proteins that are mainly involved in ER stress and energy production by mitochondria. The knockdown of TERT was shown to trigger an adaptive response to ER stress, namely unfolded protein response, which causes an increased protein regulation associated with protein folding, degradation, translation and apoptosis [129].

## 8. Therapeutic Approaches to Cure Cancer

Summarizing the data discussed above, we would like to merge the role of telomerase in cell death in the context of cancer therapy. Since telomerase activity is typically enhanced in malignant cells, many attempts have been proposed to limit tumor growth by targeting telomerase functioning [130,131]. 

The most obvious strategy in cancer therapy involving telomerase targeting is the direct telomerase inhibition. As telomerase acts similarly to other polymerases, it could be blocked by nucleotides analogues [132]. This approach showed encouraging results in reducing cancer cell growth. For example, well-studied azidothymidine (AZT), widely used for HIV treatment, positively affected survival of cancer patients [133,134,135]. Another strategy for inhibition of telomerase functioning is based on blocking the interaction between telomerase and telomeres. The stabilization of G-quadruplex DNA structure by BIBR1532 (a simple synthetic compound) and quinolones, for example, interferes in the binding of telomerase to telomeric DNA [130,136,137]. Also, TERT and TERC could be suppressed at the transcriptional level. Oligonucleotides, antisense-oligodeoxynucleotides and siRNAs, that interfere with TERT mRNA [50,138] and TERC itself, were shown to rapidly inhibit cancer cell growth and increase susceptibility to treatment [139,140]. There are several clinical trials (NCT04576156, NCT02426086) using oligonucleotide GRN163L complementarily to the template part of TERT RNA (Imetelstat), proving that approach [131,141,142]. Moreover, TERT-expressing cancer cells could be eliminated by oncolytic viruses [143]. For example, OBP-301 (Telomelysin), which is an attenuated type-5 adenovirus containing the *hTERT* promoter to regulate viral replication, can be used for this purpose. It is activated in TERT-expressing cells, but not in somatic cells in which TERT is not expressed [144,145]. OBP-301 provides clinical benefits in patients with oesophageal cancer [146]. Additionally, TERT expression can be downregulated in cancer cells under IFNa treatment. This cytokine is produced by immune cells, including T and NK cells, and usually inhibits viral replication in host cells [147]. Thus, an interesting approach is to use immune cells, such as T cells targeting TERT+ cancer cells by the recognition of TERT antigens bound to MHC [148,149,150,151]. 

Thus, downregulation of telomerase functional activity and expression level might attenuate cancer cells for treatment and diminish their proliferative potential [50,152]. However, there are several concerns about the safety of such therapy, because some normal somatic cells still express TERT and could be affected by the telomerase-directed treatment. Since TERT links different signaling pathways discussed above, its downregulation could lead to unpredictable side effects [45].

While TERT inhibition is used to suppress tumor cells, TERT overexpression can also be used to treat cancers: it is considered a prominent way to improve immune cell functioning and expand their persistence. Immune cells transferred into a recipient were found to rapidly downregulate their activity despite their accumulation within tumor sites [153]. It was shown that TERT overexpression in modified T [154] and NK cells [107,155] increases their replicative potential and overall functional activity after long-lasting cultivation. This approach might help to overcome the development of immunosuppression during cancer treatment [156,157,158]. It is known that transforming growth factor (TGFβ), common in tumor sites, represses hTERT [159] and cause immunosuppression. Additional modification of therapeutic immune cells with stably expressed TERT might help to overcome this issue. However, there is a concern about the safety of TERT-overexpressing immune cells for therapeutic treatment. Uncontrolled proliferation or even malignant transformation of these modified cells may lead to unpredictable side effects, so it is essential to irradiate or additionally modify immune cells with controlling “suicide genes” prior their infusion into patients [160,161,162,163]. 

## 9. Conclusions

TERT via interaction with a variety of intracellular signaling pathways significantly increases cell survival, including genetically modified cells with TERT overexpression, and may increase the resistance of cancer cells to therapy. However, an unambiguous mechanism for cell death suppression among cells characterized by TERT overexpression still is not completely clear. In this review, we have collected published data on the role of TERT in the regulation of cell death (Figure 4). Presumably, the complex ensemble of sophisticated intracellular interactions between telomerase catalytic subunit and other players in the main signaling pathways, NF-κB, Wnt/β-catenin and p53, corresponds to the increased survival capacity of such cells. The cross-talks occur both at the transcriptional level and through protein–protein interactions, including the modulation of the stability and functioning of various proteins. Moreover, different intracellular localizations significantly expand the spectrum of TERT functionality, allowing direct interactions with cytoplasmic and mitochondrial components. Despite these facts, most of the current data are still focused on its intra nucleus regulation and functioning, whereas there are relatively little data on the cytoplasmic and mitochondrial TERT activity. Thus, non-canonical roles of TERT need further investigation.

## Figures and Tables

**Figure 1 biomedicines-11-01091-f001:**
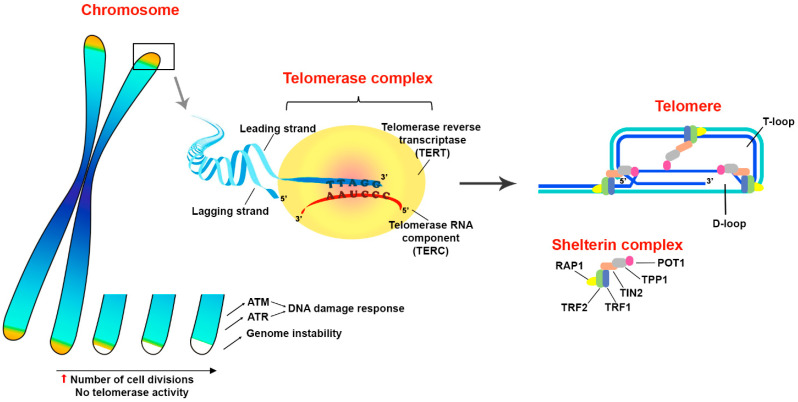
Structure and functions of telomeres. Telomeres are protective structures on the ends of eukaryotic chromosomes. The terminal DNA represented by oligonucleotide repeats and forms a secondary structure of T- and D-loops in a complex with shelterin proteins TRF1, TRF2, Rap1, TIN2, TPP1 and POT1. Telomeres prevent the induction of DNA damage response through an activation of ataxia telangiectasia mutated (ATM) and ataxia telangiectasia and Rad3-related (ATR) signaling pathways that play a principal role in the maintenance of genome integrity. Since telomeric DNA shortens in every replication cycle, because of the inability of DNA polymerase to synthesize the lagging DNA strand from 5′-end, telomere elongation is performed by a special enzyme called telomerase. Structurally, telomerase appears to be a complex ribonucleoprotein, the core part of which consists of two parts: telomerase RNA component (TERC) and telomerase reverse transcriptase (TERT). Telomerase complex adds multiple oligonucleotide sequences (5′-TTAGGGn) to chromosome ends by reverse transcription of TERC template. Thus, telomerase activity facilitates evasion of cells from death mediated by critical telomere shortage during robust proliferation. The up arrow (red) means an increase.

**Figure 2 biomedicines-11-01091-f002:**
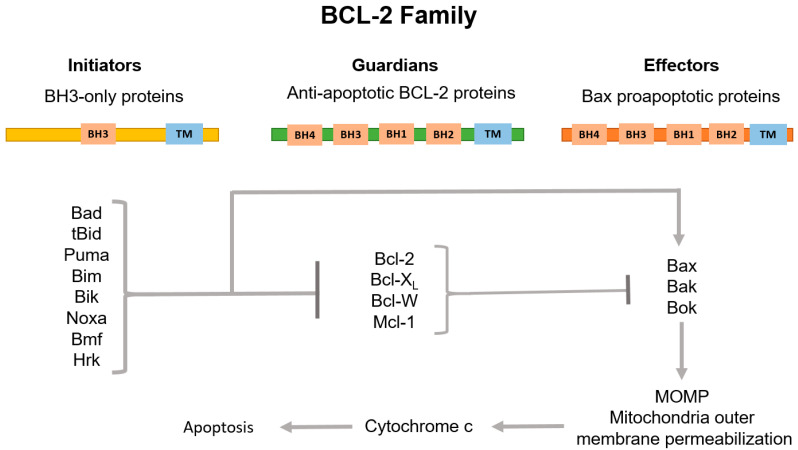
Classification of BCL-2 family proteins and their role in the regulation of the internal pathway of apoptosis. The family consists of three subgroups of proteins connected to each other by region of sequence homology, the so-called BCL-2 homology (BH) domains. BH3-only proteins, which normally possess only the BH3 domain, are activated by apoptotic stimuli to initiate signaling along the pathway. Proteins containing only BH3 interact with both BCL-2-associated X protein (Bax) and BCL-2 antagonist/killer (Bak) effectors and anti-apoptotic guardians. Guardians provide protection against apoptosis progression both by isolating proteins containing only BH3, thus inhibiting effector activation, and by direct neutralization of activated effector proteins. Upon release, the activated effectors oligomerize on the mitochondria outer membrane, which increases the permeability of this barrier. This allows the release of apoptogenic factors, primarily cytochrome c (cyt c), to the cytosol and subsequent activation of caspases.

**Figure 3 biomedicines-11-01091-f003:**
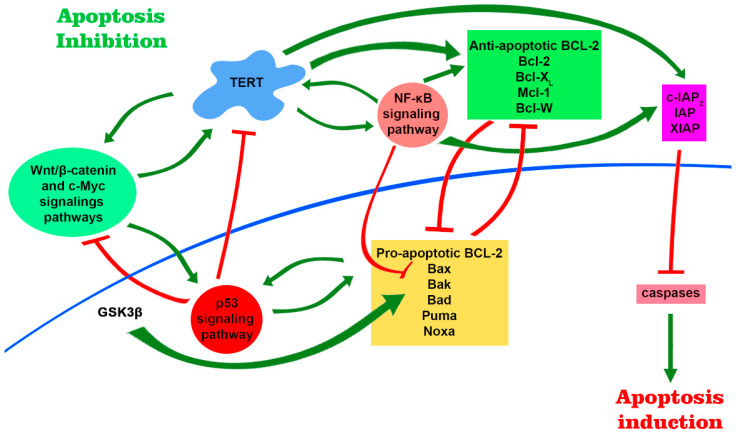
TERT is involved in major signaling pathways and participates in regulation of gene expression. There are several positive feedback loops that are formed due to mutual stimulation between TERT expression level and products of target genes of Wnt/β-catenin, c-Myc and NF-κB signaling pathways. TERT in complex with BRG1 binds to the promoter regions of the Wnt/β-catenin target genes and participates in their regulation. Further, telomerase can modulate the expression of the NF-κB signaling pathway, forming the TERT–NF-kB subunit p65 complex. This leads to an increase in the expression of pro-apoptotic genes, including Bcl-2, which is able to induce telomerase transcription and increase its functional activity. NF-κB is also able to negatively regulate the stability of p53 via E3 ubiquitin ligase MDM2. Activation of p53 leads to an increase in the level of transcription of pro-apoptotic proteins and a decrease in the level of telomerase. Furthermore, stable signaling through Wnt/β-catenin pathway leads to the accumulation of active p53, which can contribute to β-catenin degradation. The positive regulators of *TERT* include NF-κB itself, β-catenin in complex with transcription factor 4 (TCF4) and Kruppel-like factor 4 (Klf4), and c-Myc. Altogether, increased signaling of NF-κB pathway and elevated TERT levels promote cell survival by upregulation of anti-apoptotic genes, whereas p53 activation promotes apoptosis through activation of pro-apoptotic genes.

**Figure 4 biomedicines-11-01091-f004:**
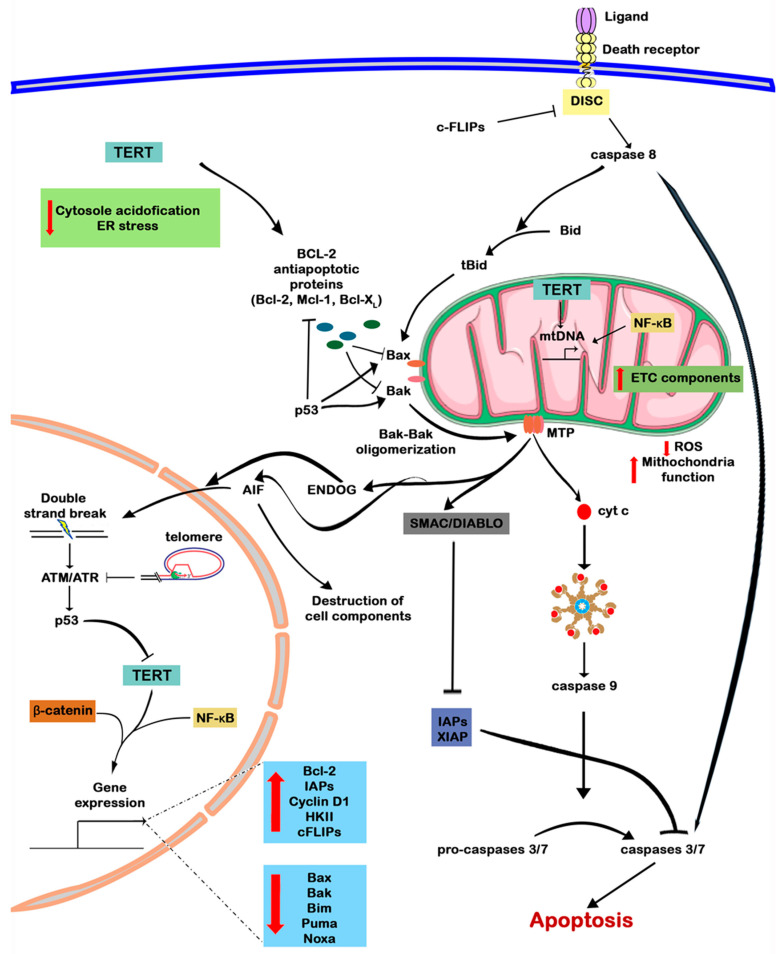
In the nucleus, TERT performs both canonical (elongates telomeric ends) and non-canonical functions (participates in the regulation of gene expression). Maintenance of a normal telomere structure suppresses ATM/ATR signaling and thus reduces the activity of p53, a negative telomerase regulator. TERT alone or together with transcription factors of NF-κB, Wnt/β-catenin signaling pathways stimulates the expression of anti-apoptotic factors, reducing the level of pro-apoptotic. In the cytoplasm, it diminishes cytosol acidification and prevents the ER stress-response progression. TERT contains a BH3-like domain which can probably bind to proteins of the BCL-2 family and regulate apoptosis. In mitochondria, telomerase reduces the level of ROS production and restores mitochondrial function. Additionally, TERT is likely to be involved in mitochondrial DNA repair and regulation of mitochondrial gene expression. Hence, BCL-2 family proteins are regulated at the transcriptional level and by protein–protein interactions with telomerase and p53. Oligomerization of proapoptotic proteins Bax and Bak leads to the formation of MOMP and the release of cyt c, SMAC/DIABLO, AIF and ENDOG from mitochondria. Cyt c in the cytoplasm induces apoptosome assembly by interacting with Apaf1 and pro-caspase 9, which leads to the activation of caspase 9 and further triggering of the cascade of caspases through activation of caspases 3 and 7 and, finally, to cell death by apoptosis. Normally, caspases are blocked by IAPs and XIAP, but SMAC/DIABLO released from mitochondria blocks them. AIF and ENDOG induce destruction of intracellular contents, along with DNA cleavage, followed by activation of ATM/ATR signaling. Apoptosis can also be induced by an external pathway from the death receptor and the formation of the DISC complex, which can be inhibited by c-FLIPs. The DISC assembly activates caspase 8, which can either directly trigger a cascade of caspases, or proteolytically cleave Bid to form tBid, which triggers the oligomerization of Bax–Bak. The up arrow (red) means increase and the down arrow (red) means decrease.

## Data Availability

Not applicable.

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
