# Peer review of "Multiple Actions of Telomerase Reverse Transcriptase in Cell Death Regulation"

_biomedicines, 2023, doi:10.3390/biomedicines11041091_

Round 1

Reviewer 1 Report

Comments and suggestions may be found in attachment

Author Response

Reviewer: Firstly, authors should change the title of manuscript in accordance to its content or to improve manuscript in accordance to its title. The title supposes that manuscript will be about nontelomere functions of telomerase components. So, it should be “Multiple actions of telomerase components in cell death regulation” and contain addition information about function of telomerase RNA not only TERT subunit. In present view manuscript is devoted to the TERT function and it should be named “Multiple actions of telomerase reverse transcriptase in cell death regulation”.

Response: We are grateful to the Reviewer for so comprehensive analysis of our review devoted to the role of telomerase reverse transcriptase in cell death regulation. We agree that the title does not properly convey the main content of the manuscript. Telomerase reverse transcriptase but not telomerase itself seems to be mainly described throughout the manuscript. So, according to your remark, we have changed the title from “Multiple actions of telomerase in cell death regulation” to “Multiple actions of telomerase reverse transcriptase in cell death regulation”.

Reviewer: The text of manuscript enriched with mistakes that complicate the perception and analysis of information. The list of mistakes may be found below.

Response: Thank you for so careful reading of our review. Your detailed list of mistakes was very useful. We have checked all recommendations listed below and corrected orthographic and logic inaccuracies.

Line 43 oligonucletid – oligonucleotide / checked

Line 47 DNA chain should be changed to DNA strand / checked

Line 72 compactification – compactization / checked

Reviewer: The first sentence of section 2.1 should be rephrased. It is necessary to point that telomerase is inactive in normal somatic cells constitutively and sometimes in specific cells it is activated to accelerate the proliferation.

Response: Thank you for your recommendation to emphasize in section 2.1 that telomerase is inactive in normal somatic cells constitutively and sometimes in specific cells it is activated to accelerate the proliferation. So, according to your remark we have rephrased whole paragraph in order to properly merge information. We also expanded the part describing telomerase activity in various somatic cells. The rephrased sentences could be found below in lines 84-101.

Reviewer: Line 97-100 rephrase please. It is not clear was do you mean in this sentence.

Response: Thank you for your notion. We have corrected the sentence (lines 97-100) to emphasize that TERT expression is associated mainly with the regulation of transcription of a group of genes coding proteins of DNA repair machinery, but without direct interaction between TERT and DNA repair machinery components itself. The rephrased sentences could be found below in lines 104-109.

Line 118 telomers – telomeres / checked

Line 121 POT – POT1 / checked

Line 126 complex protein should be changed to complex ribonucleoprotein / checked

Line 88, 129 reverse transcription from template – reverse transcription of template / checked

Reviewer: Line 129 “Thus, telomerase activity facilitates evasion from “end replication problem” for cells in a state of robust proliferation.” Please, rephrase this sentence. End replication problem is not resolved by telomerase action. But telomerase helps cells to avoid the death because of critically short telomeres.

Response: We are grateful that you noticed such logic mistake. We have rephrased this sentencе in lines 139-140.

Reviewer: Section 3. I recommend to clarify that telomerase components (but not telomerase as a complex) are involved in the regulation of different aspects of cellular fate.

Response: Thank you for your recommendation. We have expanded the heading and of section 3 and concretized the core components (TERT and TERC) of telomerase that appear to be the main figures of the section. The rephrased heading could be found in line 141. We also concretized telomerase components throughout the text.

Reviewer: Line 163-164 “The first, intrinsic apoptosis, is a pathway of cell death that develops in response to stress: such as cytokine and growth factor deprivation, ER (endoplasmic reticulum) stress, DNA damage or ROS overload [53–56].” Please clarify, intrinsic pathway of cell death does not depend on cytokines and growth factosr. This pathway regulates by internal mechanism.

Response: Obviously we agree that intrinsic apoptosis is a result of internal mechanisms of cell death regulation. However, cell stress could be caused by external factors including cytokine and growth factor deprivation (doi: 10.1083/jcb.201506118, doi:10.1126/science.280.5361.243). In review (doi: 10.1038/s41418-017-0012-4) there is “intrinsic apoptosis is a form of RCD initiated by a variety of microenvironmental perturbations including (but not limited) growth factor withdrawal.” In our review we have also noticed that intrinsic apoptosis develops in response to stress. We have rephrased lines 173-176 to clarify it.

Reviewer: Line 168-170 Please clarify. Granzyme B is not T-lymphocyte and NK-cell. It is secreted by these cells.

Thank you for your notion. We have corrected that inaccuracy. The rephrased sentences could be found in lines 178-181.

Reviewer: Line 216-217 “Thus, activation of the caspase cascade leads to the destruction of cellular components: DNA fragmentation, phosphotidilserine exposure (PS) [61].” Caspases are proteases and activation of caspase cascade results in protein proteolysis firstly and DNA fragmentation and PS exposure are the further effects of apoptosis.

Response: We are grateful for your careful reading. We have corrected that statement. The rephrased sentences could be found below in lines 228-230.

Reviewer: Line 239 “itory protein (c-FLIP) ( : c-FLIPS and c-FLIPL - act as a regulator” Please clarify.

Response: According to your recommendation, we have clarified the role of FLICE proteins in cell death regulation. The rephrased sentences could be found in lines 247-251.

Line 345 “apoptosis inhibitor (c-IAPs) and X-chromosome-associated IAP (XIAP)” These abbreviations were presented above. / corrected

Line 399 Should be “Telomerase is involved in major signaling pathways”/ corrected

Reviewer: Line 406 “transcriptional and functional activity of telomerase.” Do you mean regulation of transcription by telomerase or the regulation of transcription of telomerase?

Thank you for your notion. We have rephrased this statement to unequivocally describe Bcl-2 functions (lines 417-419).

Reviewer: Line 429-430 Mitochondria is also intracellular component. Direct interaction with mitochondria and proteins of the apoptotic pathway could not be extracellular. Please, rephrase this thesis.

The rephrased sentence could be found in lines 439-443.

Line 484 telomerese – telomerase / checked

Reviewer: Line 487-491 I recommend to expand this section and provide more detailed information about reverse transcription of tRNA in mitochondrial DNA repair.

Unfortunately, the only research (doi:10.1093/nar/gkr758) was found on the reverse transcription of tRNA in mitochondrial DNA repair. Additional information regarding reverse transcription in mitochondria could be found below in lines 543-551.

Reviewer: Line 533-535 Should be rephrase

We have rewritten the paragraph to clarify the process. The rephrased sentences could be found in lines 543-551.

Line 576 capases – caspases / checked

Reviewer 2 Report

This is a comprehensive review that was completed to a high standard. The knowledge summarized is essential to form a well-rounded understanding of telomeres and telomerase regulation in cell survival, especially the regulation pathways of cell death in human malignancies. 

The minor revision I need to point out is that the references in this manuscript are relatively old. My suggestion is to increase the proportion of articles in recent 3 or 5 years to raise more up-to-date information.

Author Response

Reviewer: The minor revision I need to point out is that the references in this manuscript are relatively old. My suggestion is to increase the proportion of articles in recent 3 or 5 years to raise more up-to-date information.

Response: Thank you for the comment. We have updated articles cited and increased the proportion of articles published in recent 3 or 5 years. In particular, we have added 36 new references and updated 16 references. The numbers of the updated articles: 1, 2, 4, 9, 11, 12, 13, 14, 17, 18, 29, 33, 34, 44, 72, 92.

Reviewer 3 Report

The review article by Palamarchuk et al., entitled “Multiple actions of telomerase in cell death regulation”, submitted for publication in Biomedicines, is a review on the role of telomerase in cell death regulation in humans. In addition to analyzing these pathways, the authors have described the various functions of telomerase and their implications in linking multiple signaling pathways. In fact, besides its early known role in telomere length regulation, telomerase is now known to regulate gene expression and also functions in stress responses and cellular metabolism. This review article is planned to be part of a special Issue of Biomedicienes entitled “Telomerase and Telomeres: Role in Health and Aging”.

            This review is well documented and well written and will undoubtfully be very useful to many readers. This is particularly remarkable as telomerase biology is extremely complex and sophisticated due to its many types of regulation and some of its yet to be iuncovered functions. Of course, these topics are of extreme importance due to their implications in cancer biology and aging.

Major comments:

 The authors in their Abstract state that “summarized the data on the role of telomerase in cell death regulation that might help to find new approaches for cancer cell treatment”, as if the subject was going to be developed in their Review. However, this statement should be toned down because there is no chapter or even paragraph devoted to cancer treatment based on the telomerase-linked pathways the authors are descrivbing. Therefore, the authors should retrieve this sentence from the Abstract. However, I propose (if the authors have a pretty good knowledge of this therapeutic aspect) that the authors include an additional chapter summarizing the various therapeutic approaches attempted to target the various pathways in which telomerase is involved in order to try to cure cancer.

Minor comments:

* line 43: “oligonucletid” should be “oligonucleotide”.

* line 64: “in the maintaining” should be “in maintaining”.

* line 99: “did not directly interacts” should be “did not directly interact”.

* line 118: “telomers” should be “telomeres”.

In fact there are many small grammatical mistakes such as those listed above. I have not noted all of them because they are too numerous and the authors should re-read carefully the entire text.

Author Response

Major comments:

Reviewer: The authors in their Abstract state that “summarized the data on the role of telomerase in cell death regulation that might help to find new approaches for cancer cell treatment”, as if the subject was going to be developed in their Review. However, this statement should be toned down because there is no chapter or even paragraph devoted to cancer treatment based on the telomerase-linked pathways the authors are descrivbing. Therefore, the authors should retrieve this sentence from the Abstract. However, I propose (if the authors have a pretty good knowledge of this therapeutic aspect) that the authors include an additional chapter summarizing the various therapeutic approaches attempted to target the various pathways in which telomerase is involved in order to try to cure cancer.

Response: We are grateful to the Reviewer for careful reading the manuscript. We have rewritten the abstract to make it reflecting the main idea of the review (lines 8-17). Also, we have added one more paragraph on the approaches in cancer treatment with the focus on telomerase (lines 528-627).

Minor comments:

Reviewer: In fact there are many small grammatical mistakes such as those listed above. I have not noted all of them because they are too numerous and the authors should re-read carefully the entire text.

Response: Thank you for your attentive reading. We carefully re-read the manuscript and checked orthographic and grammatical mistakes. Hope you find it better.

* line 43: “oligonucletid” should be “oligonucleotide”. /checked

* line 64: “in the maintaining” should be “in maintaining”. /checked

* line 99: “did not directly interacts” should be “did not directly interact”. /checked

* line 118: “telomers” should be “telomeres”. /checked

Round 2

Reviewer 1 Report

Authors performed good work to improve the manuscript and I recommend to publish it.

Reviewer 3 Report

Revision OK: comments correctly addressed by the authors and changes made accordingly.